# Genome-Wide Association Study for Maize Hybrid Performance in a Typical Breeder Population

**DOI:** 10.3390/ijms25021190

**Published:** 2024-01-18

**Authors:** Yuan Dong, Guoliang Li, Xinghua Zhang, Zhiqian Feng, Ting Li, Zhoushuai Li, Shizhong Xu, Shutu Xu, Wenxin Liu, Jiquan Xue

**Affiliations:** 1Key Laboratory of Biology and Genetic Breeding of Maize in Arid Area of Northwest Region, College of Agronomy, Northwest A&F University, Yangling 712100, China; 2National Maize Improvement Center of China, Key Laboratory of Crop Heterosis and Utilization (MOE), China Agricultural University, Beijing 100193, China; 3Leibniz Institute of Plant Genetics and Crop Plant Research (IPK) Gatersleben, 06466 Seeland, Germany; 4Department of Botany and Plant Sciences, University of California, Riverside, CA 92521, USA

**Keywords:** candidate gene, genetic effect, GWAS, hybrids, maize

## Abstract

Maize is one of the major crops that has demonstrated success in the utilization of heterosis. Developing high-yield hybrids is a crucial part of plant breeding to secure global food demand. In this study, we conducted a genome-wide association study (GWAS) for 10 agronomic traits using a typical breeder population comprised 442 single-cross hybrids by evaluating additive, dominance, and epistatic effects. A total of 49 significant single nucleotide polymorphisms (SNPs) and 69 significant pairs of epistasis were identified, explaining 26.2% to 64.3% of the phenotypic variation across the 10 traits. The enrichment of favorable genotypes is significantly correlated to the corresponding phenotype. In the confident region of the associated site, 532 protein-coding genes were discovered. Among these genes, the *Zm00001d044211* candidate gene was found to negatively regulate starch synthesis and potentially impact yield. This typical breeding population provided a valuable resource for dissecting the genetic architecture of yield-related traits. We proposed a novel mating strategy to increase the GWAS efficiency without utilizing more resources. Finally, we analyzed the enrichment of favorable alleles in the Shaan A and Shaan B groups, as well as in each inbred line. Our breeding practice led to consistent results. Not only does this study demonstrate the feasibility of GWAS in F_1_ hybrid populations, it also provides a valuable basis for further molecular biology and breeding research.

## 1. Introduction

As one of the most important food crops and industrial raw materials in the world, maize (*Zea mays* L.) plays an irreplaceable role in feed, bio-energy, and food processing, etc. [1]. Reports showed that genetic gain contributions to maize yield increases during 1930 to 2011 were 79% in the target production environment [2]. Most maize breeders develop its two complementary heterotic groups, classic examples are SS and NSS in the USA [3], and Dent and Flint in Europe [4], from which high-magnitude heterosis can be generated and the high-yield hybrid variety can easily be created through crossing between them. Meanwhile, due to the characteristics of cross pollination and C4 photosynthesis, maize has a wide range of growth adaptability and genetic diversity [5]. Along with the development of advanced sequencing technology, more and more maize inbred lines have been sequenced, and high-quality genomes have been assembled by de novo, forming the maize PAN genome [6,7,8], which highly benefits maize genomic improvement and the acceleration of research progress of complex quantitative traits. A typical example is the B73 nested association mapping (NAM) population, where researchers have dissected the genetic architecture of many complex quantitative traits and cloned many key genes, including flowering time, disease resistance, plant morphology construction, and so on [9,10,11]. The latest improvement of the maize reference genome from version 1.0 to version 5.0 has made maize a model species to clarify the genetic basis of complex agronomy traits in other species [6,12]. Subsequently, more complex traits, including drought stress, secondary metabolites, alternative splicing, and grain moisture [13,14,15,16], have been dissected with high throughput markers. Abundant molecular markers and abundant public databases provide favorable conditions to map quantitative trait loci (QTL) and explore candidate genes with genetic populations of various sizes [17,18,19].

Most QTL in maize were identified from inbred populations, such as recombinant inbred lines and doubled haploid lines. However, it is important to note that those loci may not be relevant to hybrid populations. Since the 1960s, maize has been widely planted in the form of single cross, which is the embodiment of heterosis utilization in maize. Researchers have constructed models to describe the genetic mechanism based on the F_1_ hybrids, which are valuable resources to study heterosis and realize the application of research results. Wang et al. explored the feasibility of genome-wide association studies by constructing a multiple-hybrid population (MHP) of maize F_1_ hybrids, and concluded that the mega MHP population may be widely applied in modern genetics, genomics, and breeding [20]. Zhao et al. detected 36 highly significant SNPs that affect the phenotype via additive effects, dominance effects, and genotype by environment interaction effects. These loci jointly explained 51.86–79.92% of the phenotypic variation for four quantitative traits (plant height, leaf angle, leaf length, and leaf width) in the F_1_ population [21]. Similarly, Zhang et al. identified nine important SNPs and two candidate genes associated with plant height in another F_1_ population consisting of three hundred hybrids [22]. In a study by Liu and Wang et al., 628 loci of yield-related traits were identified in 5360 F_2_ progenies from three crosses (Zheng58 × Chang7-2, B73 × Mo17, and C428 × C434). Most loci exhibited complete–incomplete dominance (the primary one) or overdominance (the secondary one) on heterozygotes, and the heterosis effects were mostly stable across different environments [23]. Additionally, Xiao et al. categorized significant loci into additive effect, dominance effect, and epistatic effect by comparing the mapping results of the hybrid phenotype, parent phenotype, and heterosis, and emphatically analyzed the developmental mechanism of the dominant effect and epistatic effect loci [24]. Similar studies on F_1_ populations have been reported in other crops. In rice, 130 significant loci associated with 38 agronomic traits were identified in a subsequent association study with 10,074 F_2_ hybrids, where most of the QTL are heterozygotes and show positive incomplete dominance [25]. In wheat, Jiang et al. analyzed the heterosis of 1604 wheat hybrids and found that the genetic variance of the epistatic effect and dominant effect was up to 64% [26]. In other related studies, scholars found that the epistatic effect plays a crucial role in the genetic architecture of the quantitative traits of maize [27,28,29]. There is significant progress in utilizing the F_1_ population to dissect the genetic architecture of complex phenotypical traits. However, especially in maize, the typical breeder population with complementary heterotic groups has seldom been emphasized. 

In contrast to inbred lines, F_1_ hybrids have various genetic effects, including additive, dominant, and all kinds of epistatic effects. To investigate the expression of these effects in hybrids, we constructed a typical breeder hybrid population including 442 F_1_ hybrids derived from the crosses between the heterotic groups Shaan A and Shaan B; crossing between them is an important and classic heterotic pattern in Northwest China. We proposed a GWAS model to dissect the genetic effects and detected candidate genes for further biological study and follow up molecular breeding. The objective is to quantify the genetic effects of different agronomic traits in hybrids and provide useful guidance for maize breeding.

## 2. Results

### 2.1. Genetic Characteristics of the Hybrid Population

All samples of the parents of the F_1_ hybrid population were genotyped using the simplified tGBS sequencing technology. Then, 19,461 high quality SNPs were filtered and used for further research in this study. Evaluating for these SNPs, the average missing frequency was 0.037, the minor allele frequency (MAF) was 0.231, the polymorphic information content (PIC) was 0.261. To analyze the heterosis pattern more clearly at the gene level, we constructed a neighbor-joining (NJ) phylogenetic tree based on the genetic distances of the inbred lines. The results showed that the Shaan B group had a much wider genetic diversity than the Shaan A group (Appendix A). This observation is consistent with our breeding strategy reported in a previous study [30]. The genotypes of all the 19,461 SNP markers of the hybrids were deduced from the genotypes of their parents. Such genotype data of the hybrids were eventually used for the genome-wide association study as described below. 

We evaluated 10 phenotypical traits for the hybrid population, grain yield (GY, Mg ha^−1^), grain moisture (GM, %), plant height (PH, cm), ear height (EH, cm), ear-leaf length (ELL, cm), ear-leaf width (ELW, cm), ear-leaf area (ELA, cm^2^), rind penetrometer resistance of the third internode above ground (RPR_TIAG, N mm^−2^), rind penetrometer resistance of the first internode under the ear (RPR_IUE, N mm^−2^), and tassel branch number (TBN). The estimated heritability of the traits ranged from 0.44 for GM to 0.82 for EH [30].

### 2.2. GWAS Model Comparison in the Hybrid Population

In order to conduct the GWAS, we proposed two models for background control; one is the simple additive background model (KinA) and the other is the full background control model (KinADE). We then evaluated the phenotypic variation explained (PVE) by the significant loci of all effects identified by the two models. Concerning the results from the additive background control model (KinA), the epistatic effects contributed to most of the phenotypic variation for all 10 traits, while the additive and dominance effects only explained a small PVE, ranging from 0 (ELL) to 13.3% (ELW) (Appendix A). 

Comparatively, under the full background control model (KinADE), the additive effects appear to contribute the greatest percentage for GY, GM, PH, EH, ELW, and ELA. For RPR_TIAG, the additive and epistatic effects contributed more or less equally. For the remaining three traits (ELL, RPR_IUE, and TBN), all three types of effects contributed to the phenotypic variation, ranging from 26.2% to 64.3% (Appendix A). Furthermore, we repeated the five-fold cross-validation one hundred times to compare the PVEs of the two models. The results showed that the KinADE model is consistently better than the KinA model for all traits (Appendix A).

By comparing the mapping results of the two models, we believe that adding all additive–dominant–epistatic effects to control the background improves the mapping results in hybrid populations. Therefore, in subsequent analyses, we only focused on the mapping results with the full background control (KinADE).

### 2.3. GWAS Results Based on the KinADE Model

There is a group of significant markers that only explain a small proportion of the phenotypic variation (PVE < 1%). Therefore, we screened markers using the threshold (PVE ≥ 1%). With this criterion, a total of 49 significant independent SNPs (main effect QTN) and 69 pairs of interaction SNPs (epistatic QTN) were detected, and these markers were subject to further analysis (Table 1, Figure 1).

Regarding independent significant SNPs, there are 3 ~ 7 main effect QTNs detected for different traits, the total PVE ranges from 12.0% (RPR_TIAG) to 59.1% (ELW), in which four main effect QTNs are identified for GY and the total PVE is 27.1% (Table 1, Figure 1). There are eleven major QTNs with high PVE (≥10%) related to six traits, specifically as one (*gySNP4*) for GY, two (*gmSNP3* and *gmSNP4*) for GM, two (*phSNP2* and *phSNP6*) for PH, two (*ehSNP1* and *ehSNP3*) for EH, two (*elwSNP6* and *elwSNP7*) for ELW, and two (*elaSNP3* and *elaSNP4*) for ELA. The biggest QTN is *elwSNP6* with an individual PVE of 22.8% for ELW (Table 1). Interestingly, all of the eleven major QTNs are additive, except *phSNP6* whose mode of effect cannot be determined clearly due to a missing homozygote in the hybrid population.

Meanwhile, there are 2 ~ 17 pairs of epistatic QTN for different traits, and the total PVE ranges from 2.9% (GM) to 36.3% (ELL). For three traits, the total PVE of epistatic QTNs are more than 20% where the contributed PVE is 36.3% for ELL, 25.5% for RPR_TIAG, and 23.6% for TBN. The contributed PVEs from epistasis are higher than the PVE from the main effect QTNs with 17.2% for ELL, 12.0% for RPR_TIAG, and 21.0% for TBN (Appendix A). There are three epistatic QTNs whose individual PVEs are above 5% with 6.8% from *ellEpi5* (chr3:121297477 × chr3:139767124) for ELL, 6.4% from *tiagEpi9* (chr3:82338753 × chr7:18384534) for RPR_TIAG, and 6.2% from *tbnEpi1* (chr1:12106656 × chr8:165204139) for TBN.

In general, significant loci explain from 26.2% to 64.3% of the phenotypic variation for the 10 traits in the hybrid population. Among them, the additive effect contributes the most for GY (25.0%), GM (42.8%), ELW (56.7%), PH (33.7%), EH (37.9%), and ELA (44.9%). The epistatic effect contributes the most for ELL (36.3%), RRP_ TIAG (25.5%), and TBN (23.6%). The PVEs of the additive/dominance effect (11.7%), the dominance effect (7.6%), and the epistatic effect (6.9%) are much the same for trait RPR_IUE (Appendix A).

### 2.4. Allele Frequencies of QTNs and Favorable Genotype Accumulation Effects for the 10 Traits

The allele frequency distribution of each QTN provides information for the selection status of breeding, and the favorable genotype identification can be used for future crop improvement. The accumulation effect of favorable genotypes can be used to validate the QTNs for each trait. 

The allele frequency distribution is illustrated in the left diagram and the relevant genotype effect distribution is shown in the right boxplots (Figure 2 and Appendix A). Almost all the alleles are unevenly distributed in both the Shaan A and Shaan B groups; especially, only allele “*T*” of *iueSNP3* appears in Shaan A (Figure 2C) and only allele “*C*” of *ellSNP3* appears in Shaan A (Appendix A). The uneven allele frequency distributions imply that these QTNs have been subject to selection in the breeding process. For some QTNs, the major alleles of the two groups are complementary. For example, for *gySNP1*, the major allele of Shaan A and Shaan B are “*A*” and “*T*”, respectively. For most of QTNs, however, the major alleles in both groups are the same, as shown by *ehSNP1*, *iueSNP1*, *tbnSNP1*, *gmSNP3*, etc. 

Boxplots of the genotype effect of each QTN are highly consistent with the mapping results, that is, for all the additive effect QTNs, the phenotypes of individuals with heterozygotes tend to be in the middle of the two homozygotes. For loci with complete dominance effects, such as *iueSNP2*, *tbnSNP5*, *tbnSNP6*, *elwSNP1*, *tiagSNP1*, etc., the phenotypic values of the heterozygotes are higher or lower than the two homozygous genotypes. To be cautious, all loci with Add/Dom effects only appear in one form of the homozygote genotype in the F_1_ population, which makes it impossible to determine whether the locus is Add, Dom, or both (Figure 2 and Appendix A). 

From the phenotypic prediction of every genotype of all the main effect QTNs and epistatic QTNs (Appendix A), we determined the favorable genotype for each QTN. To examine the cumulative effect of the favorite genotype of the identified QTN for each trait, we fitted the regression lines between the predicted phenotypic value and the cumulative number of favorable genotypes, showing that all the linear regressions were statistically significant (*p* < 0.001). Regarding the coefficients of determination (R^2^), four traits (GM, PH, EH, and ELW) had relatively high R^2^ values, ranging from 0.299 to 0.404, and three traits (GY, ELA, TBN) had intermediate R^2^ values with 0.187 for GY, 0.189 for ELA, and 0.176 for TBN (Figure 3). These results further validate the mapping results.

### 2.5. Candidate Gene Identification

According to the linkage disequilibrium (LD) decay distance (150 kb) and the public expression data of various tissues from the MaizeGDB database (https://maizegdb.org/, accessed on 6 June 2023), a total of 532 protein-coding genes show moderate expression (FPKM > 10), including 40 for GY, 35 for GM, 64 for PH, 27 for EH, 109 for ELL, 36 for ELW, 67 for ELA, 78 for RPR_ TIAG, 24 for RPR_ IUE, and 59 for TBN. Among them, seven candidate genes are common for two groups of traits (Appendix A).

Among those candidate genes, *Zm00001d044211* contains a SNP (C/G) associated with *gySNP3* and it encodes phosphoglucomutase and participates in the metabolism of glucose, starch, and sucrose by interacting with phi1, gpm274, csu815, and other proteins (Figure 4A–C). As an additive effect gene, the phenotype of genotype CG is between the phenotypes of genotypes CC and GG (Figure 4D). In the hybrid (CL11 × NG5) [28], the expression level of *Zm00001d044211* at the spikelet and the florescence stages is in the intermediate level of its parents (Figure 4E). Our transcriptome data of kernels at different stages [29] shows that the expression levels of *Zm00001d044211* in two low-yield inbred lines (KA225 and KB182) are significantly higher than the two inbred lines (KA105 and PH6WC) with high yield, especially at 43–56 days after pollination (Figure 4F). This observation indicates that the gene may affect yield by negatively regulating starch synthesis.

Moreover, some genes related to cell division, growth, and differentiation, may also play an important role in yield development, such as *Zm00001d032146* (calcineurin subunit B), *Zm00001d011525* (mitogen-activated protein kinase 8-like), *Zm00001d017863* (protein-tyrosine-phosphatase), *Zm00001d032166* (protein-enhanced disease resistance 2), *Zm00001d044212* (auxin-induced protein pcnt115). For GM, gene *Zm00001d038218* (cytochrome C) may affect grain moisture by regulating cell apoptosis, while *Zm00001d001800* and *Zm00001d001802* (GDSL esterase/lipase CPRD49) may respond to ethylene-mediated metabolism. Meanwhile, the PH-related genes *Zm00001d037271* and *Zm00001d045480* encode microtubule-associated proteins, and the candidate genes (*Zm00001d011901*, *Zm00001d048373*, *Zm00001d017435, Zm00001d018030*, *Zm00001d021084*, and *Zm00001d053782*) related to leaf traits encode pigments or proteins related to photosynthesis. For RPR_TIAG, two linked candidate genes *Zm00001d038982* (Golgi-apparatus-related) and *Zm00001d038984* (photosynthesis-related) may play an important role in the construction of the plant stem (Appendix A). Moreover, we also observed related functional genes in the candidate genes near the epistatic sites. For ELL, there are two pairs of *ellEpi1*/*ellEpi2* with common sites, while the candidate genes of the three interaction sites all have genes related to ubiquitin synthesis (*Zm00001d017252*/*Zm00001d043565*/*Zm00001d046890*) (Appendix A). The identification of these candidate genes can provide a reference for subsequent functional verification and molecular marker development for marker assistant selection breeding.

According to the characters, the ten traits are classified into four groups to conduct the GO enrichment analysis. The four groups are yield-related (GY and GM), plant architecture (PH, EH, and TBN), ear-leaf (ELL, ELW, and ELA), and rind penetration strength (RPR_TIAG and RPR_IUE). The results show that, for yield-related traits, the expression is enriched in the organelle membrane, with the synthesis and metabolism of hypoxanthine and female pregnancy. For plant architecture, the organophosphate metabolic process, the organophosphate biosynthetic process, and the cellular component biogenesis may participate in the construction of PH and EH. For the ear-leaf traits, the expressions are enriched in the biological process of organic nitrogen and carotene, while a noticeably related item for the rind penetration strength is the cell cortex region in the cell component (Figure 5).

## 3. Discussion

### 3.1. The Main Genetic Effects of Different Traits with Existential Discrepancy

Scientists and plant breeders have conducted fundamental research on the genetic composition of yield. However, due to differences in experimental design, type of population, and complexity of the traits, the results from various studies are often different. Sometimes, the conclusions are opposite. However, one consensus unarguably exists, which is that all effects play an irreplaceable role in grain yield development. For trait GM, researchers generally believe that the additive effect plays a primary role, although the additive × additive interaction effect also exists [31,32,33]. In terms of plant structure variation, previous reports showed that plant height and ear height are primarily controlled by additive and epistatic effects, while leaf-related traits are mainly controlled by multiple genes with small effects and very little epistatic effects [10,21]. For stalk strength, previous studies have shown that it is a complex quantitative trait controlled by multiple genes with small effects [34,35]. Although it has been reported that the additive effect has a great influence on the phenotypic variation of tassel branch number, the development of tassel branch is controlled by multiple genes, and its potential molecular mechanism still needs to be further studied [36,37].

In this study, we concluded that GY, GM, PH, and EH are mainly controlled by the additive effects. This is not only consistent with the results reported previously, but also consistent with the goal of our breeding program, that is, to take advantage of the additive beneficial genetic effects so that they can be passed from parents to the hybrid progeny, simultaneously maintaining a certain level of heterosis. On the other hand, the results of our study are also different from previous studies, in that the epistatic effects are important for traits ELL and RPR_TIAG, both the additive and epistatic effects play a major role in traits ELW and ELA, and dominance and epistatic effects jointly contribute to the variation of traits RPR_IUE and TBN. 

### 3.2. Advantages of Genetic Analysis in Hybrid Populations 

In the past, most GWAS were conducted in natural populations or progeny populations derived from a small number of parents. The results cannot be directly applied to hybrid populations [20]. More evidence has pointed to the complementary advantages of harmful alleles and the heterozygosity of individual loci contributing to the formation of phenotypes [38,39,40]. Many studies concluded that epistatic effects are underestimated due to the limited population sizes [9,24,41]. The effects of dominance and epistasis on traits now can be understood intuitively in hybrid populations.

Although additive effects are still the most important effect for many traits, dominance and epistasis cannot be ignored. For example, for traits GY and EH, although dominance effects only explain a small proportion of the phenotypic variance, they can increase differences in phenotypes more substantially than the additive effects. Particularly, dominances are the primary effects for traits RPR_IUE and TBN. Combining the 14-day seedling transcriptomic data of B73, Mo17, and B73 × Mo17 [42], 102 out of 197 candidate genes for four traits were considered to be affected by expression levels during this period (PH, EH, RPR_TIAG, and RPR_IUE). Among the 102 genes, 37 (36.3%) show positive or negative overdominance in F_1_, of which *Zm00001d033132* (*iueSNP1*, chlorophyll a-b binding protein), *Zm00001d022237* (*phEpi6-2*, RNA binding protein 1), *Zm00001d011826* (*tiagEpi2-2,* NAD(P)H-quinone oxidoreductase subunit O chloroplastic), and *Zm00001d038984* (*tiagSNP1*, photosystem I reaction center subunit VI, chloroplastic precursor) have high expression levels (FPKM > 100) (Appendix A). In another transcriptome data from spikelets and florets of two lines (CL11 and NG5), and their hybrid (HYB) [43], 44 and 34 genes of the 75 GY and GM-related candidate genes, respectively, are expressed in spikelets and florets, of which *Zm00001d001808* (*mSNP1,* bystin) and *Zm00001d002326* (*gyEpi2-1,* gamma-aminobutyrate transaminase 1) are overdominant for spikelets, but no genes had overdominance expression for florets, while only six genes had the same expression level as their parents (Appendix A). These results indicate that the dominance effect cannot be ignored when analyzing the genetic structure in maize hybrids.

For epistatic effects, we located a great number of epistatic interaction sites, and the contribution of these sites to the phenotypic variation is often very low. However, we have good reasons to believe that these minor effect sites play a key role in the development of quantitative traits [26]. 

### 3.3. The Genetic Distance of Core Tester Lines within the Heterotic Group Influences Little to GWAS Using the Inter-Heterotic-Group Hybrid Population

The inter-heterotic-group hybrid population was formed with four and five selected core testers from the Shaan A and Shaan B groups, respectively. It is interesting to know whether the genetic distances of the core tester lines within the heterotic group influence the power of the GWAS. We proposed a two-sub-sampling strategy to answer the question: (1) One subsample consists of hybrids from the crosses of tester lines (A008, A009, A021) and (B018, B031, B054), where the three lines within each heterotic group are very close (Figure 6A). This is called the close strategy; (2) The other subsample consists of hybrids from the crosses of core tester lines (A009, A019, A021) and (B031, B110, B137), where the tester lines within each heterotic group are distantly related (Figure 6A). This is called the far strategy. The mapping efficiencies of the two were compared for six traits: GY, GM, PH, EH, ELA, and RPR_TIAG. In terms of PVE for trait GY, the close strategy has a highest value of 53.5% and the far strategy has a lowest value of 21.4%. For trait EH, however, the close strategy has a lowest value of 9.9% and the far strategy has a middle value of 42.1% with a highest value of 61.4% overall. For the remaining traits, not many differences were observed among the three types of mapping populations (Figure 6B). In terms of the number of positive QTNs, the close strategy appears to be greater than the far strategy for both the additive (Figure 6C) and the dominance effects (Figure 6D), but less than the original total population. This may be due to the decreased population sizes in the subsamples. Interestingly, the number of positive epistatic effects from both the close strategy and the far strategy are greater than the number in the original full population (Figure 6E), although the subsamples are much smaller than the original population. Considering the GWAS efficiencies for six traits, the genetic distances between the core tester lines within the heterotic group had little influence on the GWAS power. We conclude that breeders and geneticists can select the core tester lines based on breeding experience or operational convenience to construct hybrid populations for the GWAS. 

### 3.4. The Inter-Heterotic-Group Hybrid Population Can Be Optimized by Increasing the Number of Core Tester Lines without Increasing Resource Cost 

The ideal hybrid population should include a complete set of genotypes for each locus, that is two homozygotes and one heterozygote. However, there are only two genotypes at many of the positive QTNs, such as *gySNP2* and *ehSNP3.* The reason is that the locus is monomorphic within either the Shaan A or Shaan B core tester lines. Concerning each locus, if only two genotypes appear in the hybrid population (one homozygote and one heterozygote), it is hard to distinguish the additive and the dominance effects thoroughly. It is also impossible to separate different epistatic effects. Since one homozygote is missing in the hybrid population for such locus, we call this type of locus a one-genotype-missing-site (OGMS). In this study, we only selected four and five core tester lines from the Shaan A or Shaan B group, respectively. Therefore, the chance of either group being monomorphic is extremely high considering the small numbers of germplasms. 

We now propose an optimal strategy of core test line selection while keeping the population size of the hybrids constant. We can randomly select part of the core tester lines to cross with the counterparts. Figure 7A shows the current cross strategy, with four core tester lines from the Shaan A group and five core tester lines from the Shaan B group. In Figure 7B, the core tester lines from both Shaan A and Shaan B are doubled, but just half of them are randomly selected to cross with their counterparts. Figure 7C shows that the core tester lines from both Shaan A and Shaan B are tripled, but just one third of them are randomly selected to cross with their counterparts. We counted that the number of OGMSs in the current strategy is 6150 through 19,461 SNP markers. Furthermore, we obtained the median value of OGMS numbers, which are 4370 and 3993 for the doubled and tripled strategy, respectively. Therefore, compared with the current strategy, the number of OGMSs can be decreased by 28.9% and 35.1% for the doubled and tripled strategy, respectively.

### 3.5. Heterozygosity Contributes Significantly to Hybrid Yield

For trait GY, the additive, dominance, and epistatic QTNs contribute in total to 30.4% of the PVE, where the additive effects contribute the most at 25.0% (Appendix A). It seems that heterozygosity contributes little to hybrid yield. We evaluated the genotypes for the top 10% highest yield (labeled as TOP) and the bottom 10% yield (labeled as BOT). The number of cumulative favorable genotypes and the number of heterozygotes were counted for each hybrid. The distributions of the main effects and the epistatic effects are shown in Figure 8A and 8B, respectively. From the boxplots and the multiple comparisons, we can see that the number of heterozygotes in the TOP group is significantly higher than that in the BOT group, which indicates that heterozygosity contributes significantly to the hybrid yield.

We can continuously increase the yield of the inter-heterotic-group hybrids in two ways. One way is to introduce new favorite genes into the breeding population. The other is to arrange and solidify those favorite genes into paired groups complementarily, which creates the highest number of heterozygotes in the hybrid population. 

### 3.6. Using Genomically Predicted Phenotypes Does Not Increase the Power of the GWAS 

There were several studies that first predicted values of the untested hybrids and then included the imputed phenotypes of the untested hybrids to perform the GWAS, hoping to increase the QTN detection power by increasing population sizes [24,44]. Based on the existing research [30], we extended the current population size by the genome selection method Reproducing Kernel Hilbert Space (RKHS) and identified significant sites for additive and dominance effects in the predicted population using the KinADE model (Figure 9). In the predicted population, except for RPR_TIAG, the overall trend was basically consistent with the positioning results of the current population (R^2^ 0.80–0.94) for the additive effects. However, for the dominance effects, the significance of a large number of SNPs was irregularly increased, and the overall trend was significantly different from the results of the current population (R^2^ 0.12–0.24) (Figure 9). However, this practice does not provide any additional information; it merely copied the genetic information of the original population multiple times, which superficially increased the peak values of the significance tests. Therefore, the results from the imputed phenotypes for the GWAS are misleading. The only right way to improve the power and resolution of the GWAS is to increase the number of parents and the number of hybrids.

The sizes of the inter-heterotic-group hybrid population can be increased through historical breeding data. In general, it is not easy to increase the population size of tested hybrids in a single year. Plant breeders often use the elite lines to produce more hybrids in multiple test years. Since the constant controls are within multiple-year trials, we can combine these data to draw the best linear unbiased estimation (BLUE) of the tested hybrids in different years, and thus significantly increase the population size. Using historical breeding data for GWAS can: (1) increase population sizes; (2) improve resource abundance; and (3) consequently reduce the influence of genotype-by-environment interactions (G × E) on the estimation of genotypic values, leading to throughput upgrading, resolution improving, and power increasing for GWAS.

### 3.7. Current Research on Candidate Genes

By comparing the previous results, a great number of candidate genes identified in this study have been reported. For example, among the candidate genes related to GY, *Zm00001d044201*/*Zm00001d044203* have been reported to be related to kernel row number per ear [45]. Furthermore, another five candidate genes including *Zm00001d002326*, *Zm00001d002330*, *Zm00001d011534*, *Zm00001d032162*, and *Zm00001d044213* were found to play an important role in the endosperm development of maize [46]. One particular gene (*Zm00001d044211*) encoding phosphoglucomutase (PGM) has been proven to play a crucial role in starch biosynthesis. A significant change in starch content has been observed in sweet potato tubers when overexpressing *PGM* [47,48], indicating the potential involvement of this gene in maize kernel development. Of course, other candidate genes also deserve to be highlighted due to their participation in grain formation, such as GY-related genes *Zm00001d011525*, *Zm00001d017863*, and *Zm00001d029512*, which encode proteins MPK8 (pollen development and pollen tube formation), IBR5 (a dual-specificity phosphatase-like protein-modulating auxin and abscisic acid), and DOF36 (positively regulates starch synthesis), respectively [49,50,51]. 

For GM, the candidate gene *Zm00001d001809* has been found to be associated with the water content of shoots and leaves, and the total solid sugar content [52]. In addition, protein CPRD49 is encoded by genes *Zm00001d001800* and *Zm00001d001802*, and belongs to the GDSL esterase/lipase family, which has been reported to be related to the development of rice pollen exine and the regulation of ethylene, while CPRD49 itself has also been reported to be closely related to flower development in Chinese cabbage (*Brassica rapa* L.) [53,54,55]. Furthermore, a RPR_TIAG-related gene, *Zm00001d018625*, that encodes the same protein family with the hardness of soybean seed coat control gene GmHs1-1, has been reported to significantly contribute to the utilization of phosphorus [56,57]. A TBN-related gene, *Zm00001d027722* (CNR13), which is a member of the CNR gene family, has been shown to play a key role in yield, plant size, plant morphology, and heterosis, among which CNR13 has a high expression level in the tassel of maize [58]. More candidate genes have been reported in previous studies [28,59,60,61]. Those candidate genes provide valuable references and promising goals for gene cloning and marker assistant selection breeding.

### 3.8. Predominant Alleles among Populations

Meanwhile, to explore whether there is a gene selection preference between the two populations in the process of breeding selection, we discussed the distribution of favorable alleles of significant loci in the two heterotic groups. Considering our breeding target, we regarded the allele with a lower phenotypic level as the favorable allele for traits GM, EH, and TBN, while the other seven traits are opposite. The results show that the favorable genotypes of these ten traits do not show an obvious enrichment trend in either the Shaan A group or Shaan B group. On the other hand, the indifferent frequency of favorable genotypes in group A and group B of a single locus may be the result of our population selection process to improve adaptability, while the differences in favorable genotypes were due to the different requirements of the male parent group and female parent group. However, in general, there were some differences between Shaan A and Shaan B in the selection of traits. For example, the proportion of materials with favorable alleles in group A was higher than group B for GY, RPR_TIAG, and RPR_IUE, while the proportion in group B was higher for trait GM (Appendix A). When we calculated the proportion of favorable alleles of each material in different traits to the total number of alleles, we were surprised to find that A008(KA105) enriched the highest proportion of favorable alleles, and this material happened to be our core elite line, which cultivated a lot of excellent varieties in the past few years (five approved nationally and thirteen approved by the Shaanxi province in China), such as Shaandan650, Shaandan660, Shaandan620, etc. (Appendix A). This confirms not only the effectiveness of our positioning, but also the effectiveness of our breeding process. In addition, A019, also as the core material, has great potential in increasing yield, while B102 has great potential in reducing grain moisture (Appendix A).

## 4. Materials and Methods

### 4.1. Materials and Field Experiment Design

A total of 486 hybrid seeds were generated from crosses between 30 inbreeds of the Shaan A group and 89 inbreeds of the Shaan B group with a partial factorial design in 2016 at the experimental station in Hainan, China. In the next year, all crosses were planted in two locations in the Shaanxi province of China, Yangling (34° 16′ N, 108° 40′ E) and Yulin (38° 30′ N, 109° 77′ E). Field trials were carried out using an incomplete block design with two replications. To correct the block effect, four commercial hybrids were used as checks, making a total of 490 hybrids that were field evaluated. Since part of the parental lines were not genotyped, only 113 inbred lines and their 442 hybrids were used in the study. Details of the genetic materials and phenotypic data collection have been described in a previous report [30].

### 4.2. Genetic Characteristics Analysis

Leaves of each parent of the hybrids were collected at the 3-leaf stage and DNA samples were extracted by a modified CTAB method [62]. All samples were genotyped using the tGBS technology (Dara2bio; LLC, Ames, IA, USA) [63]. In the end, a total of 19,461 high-quality SNPs were filtered from 48,415 SNPs with a minor allele frequency (MAF) smaller than 0.05 and missing ratio greater than 0.10. We imputed the missing values with the LD KNNi method of the Tassel software package (V 5.2.70). From the filled genotype data, we evaluated the distribution of the SNPs on the genome with the CMplot package in R (4.04) and drew the phylogenetic tree of the parents with the neighbor-joining (NJ) method of the Mega7 (V 7.0.26) package [64].

### 4.3. Genome-Wide Association Studies in Hybrid Populations

The additive (Add) and dominance (Dom) effects of each locus and pairwise interaction effects (epistatic, EPI) can be estimated in genome-wide association studies (GWAS). The baseline model used for genome-wide association studies is the mixed linear model [65] which has become a standard method for GWAS. In order to control the polygenic background effects (the structure of multiple levels of cryptic relatedness), marker-derived kinship matrices were used to control the polygenic background [66]. The mixed linear model is: (1)y=Xβ+Mα+g+e
where y is a vector of the adjusted phenotypic values of the hybrids, *X*β represents fixed effects other than the SNP under investigation. Since the phenotypes have been adjusted by macroenvironmental effects, *X*β is simply the intercept. The SNP effect is denoted by α that is treated as fixed. Depending on the type of model, α can be the main effect of a SNP (additive or dominance effect) or an epistatic effect (additive-by-additive, additive-by-dominance, dominance-by-additive, or dominance-by-dominance epistatic) for a pair of SNPs, and M is a design matrix (vector) corresponding to fixed effect α. For each tested SNP, the additive design matrix (Z) is coded as −1, 0, and 1 for the first homozygote, the heterozygote, and the other homozygote, respectively; the dominance design matrix (W) is coded as 0 for homozygote and 1 for heterozygote. Each column of the epistatic design matrix is generated by a direct product [66] of two columns of the main effect design matrix. In model (1), g denotes a vector of polygenic effects and e is a vector of residuals assumed to be distributed as $e\sim N\left(0,I\sigma_{e}^{2}\right)$.

For models that only capture additive effects, $g=g_{a}$, and $g_{a}\;\sim N\left(0,K_{a}\sigma_{a}^{2}\right)$, where $K_{a}$ is the additive kinship matrix. The additive version of model (1) is:(2)$$y=X\beta +M\alpha +g_{a}+e$$
which has been incorporated in the rrBLUP package (v4.0.4).

We modified model (1) to incorporate all model effects, including the main effects and epistatic effects to control the polygenic background [26,66], as shown below:(3)$$y=X\beta +M\alpha +g_{a}+g_{d}+g_{aa}+g_{ad}+g_{da}+g_{dd}+e$$
where $g_{a}\sim N\left(0,K_{a}\sigma_{a}^{2}\right)$, $g_{d}\sim N\left(0,K_{d}\sigma_{d}^{2}\right)$, $g_{aa}\sim N\left(0,K_{aa}\sigma_{aa}^{2}\right)$, $g_{ad}\sim N\left(0,K_{ad}\sigma_{ad}^{2}\right)$, $g_{da}\sim N\left(0,K_{da}\sigma_{da}^{2}\right)$, and $g_{dd}\sim N\left(0,K_{dd}\sigma_{dd}^{2}\right)$, and $K_{d}$, $K_{aa}$, $K_{ad}$, $K_{da}$, and $K_{dd}$ are the corresponding genomic kinship matrices calculated from genome-wide markers [66].

When we tested the epistatic effect between two loci using model (3), the model included the two main effects and the epistatic effect between the two loci, as shown below:(4)$$y=X\beta +M_{i}\alpha_{i}+M_{j}\alpha_{j}+\left(M_{i}{}^{\circ}M_{j}\right)\alpha +g_{a}+g_{d}+g_{aa}+g_{ad}+g_{da}+g_{dd}+e$$
where $\alpha_{i}$ and $\alpha_{j}$ are the main effects and α (without the subscripts) is the epistatic effect between the two loci. Models (3) and (4) were implemented with the modified code developed by Jiang et al. (2017) [26].

Using the method proposed by Gao, et al. [67], we obtained 883 independent significant markers (PCA = 99.5%) from a total of 19,461 markers. Therefore, the thresholds of the dominant effects and additive effects were set at 3.95 (= 0.10/883) and the threshold for epistatic effects at 6.59 (= 0.10/(883 × (883 − 1)/2)). Due to the many interaction sites for trait TBN, we set a more stringent threshold at 7.59 (= 0.01/(883 × (883 − 1)/2)) for the epistasis effect of this trait. Based on the LD decay distance (150kb) in the parental population, we chose R^2^ > 0.2 as the criterion to decide the most significant SNPs.

### 4.4. Candidate Gene Selection and Annotation of Associated SNPs

Through comparing with the database of Genome B73_RefGenV4 on the website of MaizeGDB (https://www.maizegdb.org/, accessed on 6 June 2023), we defined genes within the 150 kb (the LD decay distance) range surrounding the association site as candidate genes. We used the MaizeGDB (https://www.maizegdb.org/, accessed on 13 June 2023) and NCBI (https://www.ncbi.nlm.nih.gov/, accessed on 13 June 2023) databases to annotate the functions of the identified candidate genes and annotate the functions of the proteins coded by the candidate genes using the online database InterPro (https://www.ebi.ac.uk/interpro/, accessed on 25 June 2023). For the GO enrichment analysis, we used the tools in the Gene Denovo website (https://www.omicshare.com/tools/, accessed on 10 June 2023). The expression levels of the candidate genes were obtained from the transcriptome data sets in NCBI Gene Expression Omnibus, including the transcriptome data of kernels at different stages with an accession number GSE15881 [68], the transcriptome data of hybrid seedlings with an accession number GSE155947 [42], and the transcriptome data of maize spikelets and florets with an accession number SRP066518 [43].

## 5. Conclusions

This study demonstrates the effectiveness of genetic analysis in a typical breeder hybrid population through a GWAS of 10 quantitative traits in maize. Comparing with the conventional populations of inbred maize, hybrid populations offer greater insights into the genetic architecture of agronomy traits, including specific effects such as dominance and epistatic effects. Furthermore, we identified a potential gene, *Zm00001d044211*, and other candidate genes for grain yield-related traits. The results provide a novel theoretical and molecular guidance for maize genetic studies and breeding for agronomy traits. Overall, this study highlights the importance of genetic analysis in hybrid populations and offers valuable insights in hybrid breeding. 

## Figures and Tables

**Figure 1 ijms-25-01190-f001:**
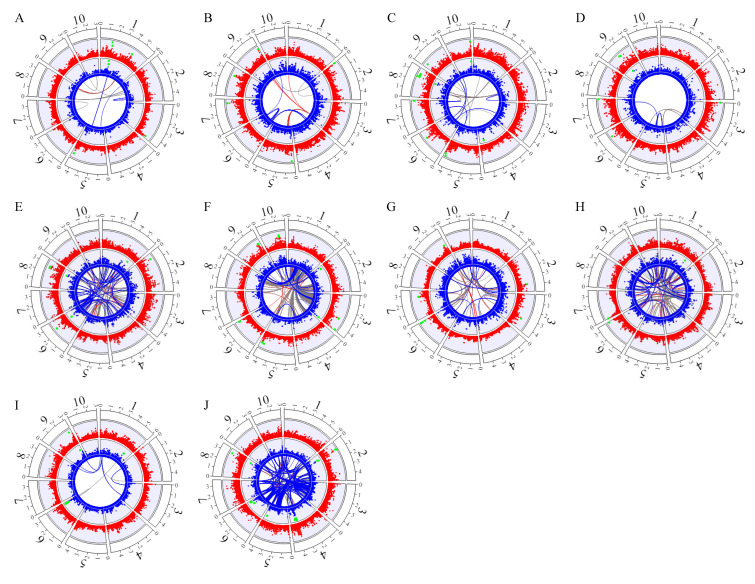
Genome-wide association studies for ten traits based on model KinADE. The red points refer to the −log_10_ (*p*-values) of the additive effects, and the blue points refer to the −log_10_ (*p*-values) of the dominance effects. The green dots mark the positive QTN (−log_10_ (*p*-values) > 3.95). Links in the center of the circle represent positive biallelic interactions between SNP markers; red lines reflect additive-by-additive, blue lines additive-by-dominance, and gray lines dominance-by-dominance interactions. (**A**) GY, (**B**) GM, (**C**) PH, (**D**) EH, (**E**) ELL, (**F**) ELW, (**G**) ELA, (**H**) RPR_TIAG, (**I**) RPR_IUE, (**J**) TBN.

**Figure 2 ijms-25-01190-f002:**
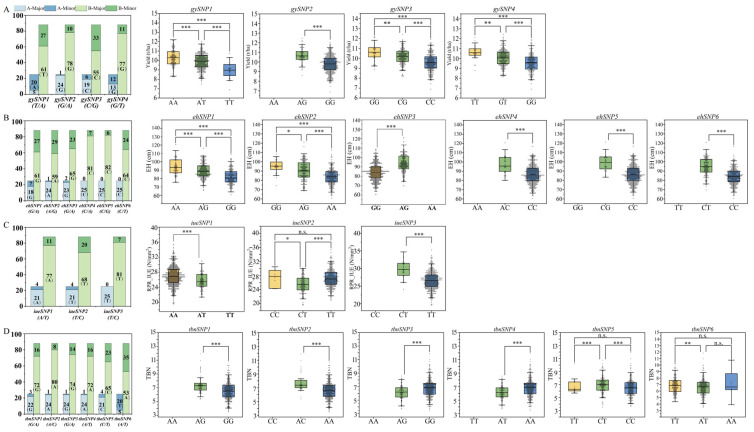
Allele frequency and phenotypic information of different genotypes of each main effect QTN for GY (**A**), EH (**B**), RPR_IUE (**C**), and TBN (**D**). A-Major and A-Minor refer to the globe major and globe minor allele distribution within the Shaan A group, respectively; B-Major and B-Minor refer to the globe major and globe minor allele distribution within the Shaan B group, respectively. n.s., *, **, and *** mean significant at *p* > 0.05, *p* ≤ 0.05, *p* ≤ 0.01, and *p* ≤ 0.001, respectively.

**Figure 3 ijms-25-01190-f003:**
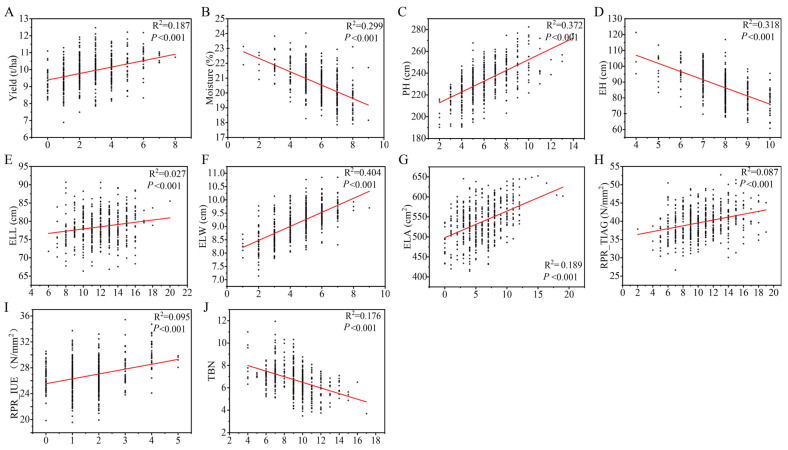
Correlation between phenotypes and the number of favorable genotypes for 10 traits. The red line of each panel refers to the fitted regression line, R^2^ is the coefficient of determination (the strength of association), *p*-value is the significance level of the linear regression. (**A**) GY, (**B**) GM, (**C**) PH, (**D**) EH, (**E**) ELL, (**F**) ELW, (**G**) ELA, (**H**) RPR_TIAG, (**I**) RPR_IUE, and (**J**) TBN.

**Figure 4 ijms-25-01190-f004:**
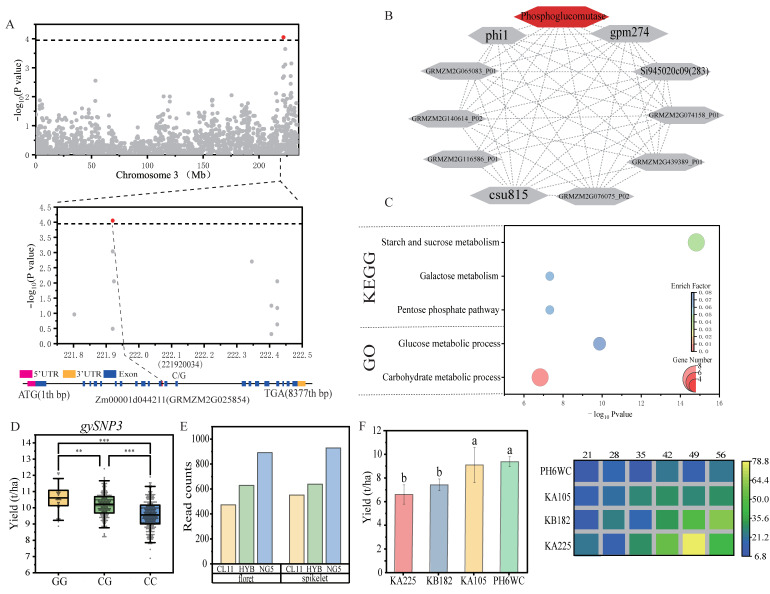
Multi-omic identification of candidate gene *Zm00001d044211*. (**A**) Structure and mutation sites of the GY-related gene (*Zm00001d044211*) on chromosome 3; (**B**) Protein–protein interaction network related to the candidate gene. Red indicates the protein encoded by the candidate gene; (**C**) Biological functions of candidate genes and the interaction networks; (**D**) Phenotypic level of different genotypes in the hybrid population. ** and *** mean significance levels at 0.01 and 0.001, respectively; (**E**) Transcription level of gene *Zm00001d044211* at the floret and spikelet stages in two inbred lines and their hybrid; (**F**) The bar plot shows the yield of the four inbred lines, and the heat map shows the expression level of gene *Zm00001d044211* from the grains of the four inbred lines at different stages. Different letters show significant difference at *α* = 0.05 for grain yield.

**Figure 5 ijms-25-01190-f005:**
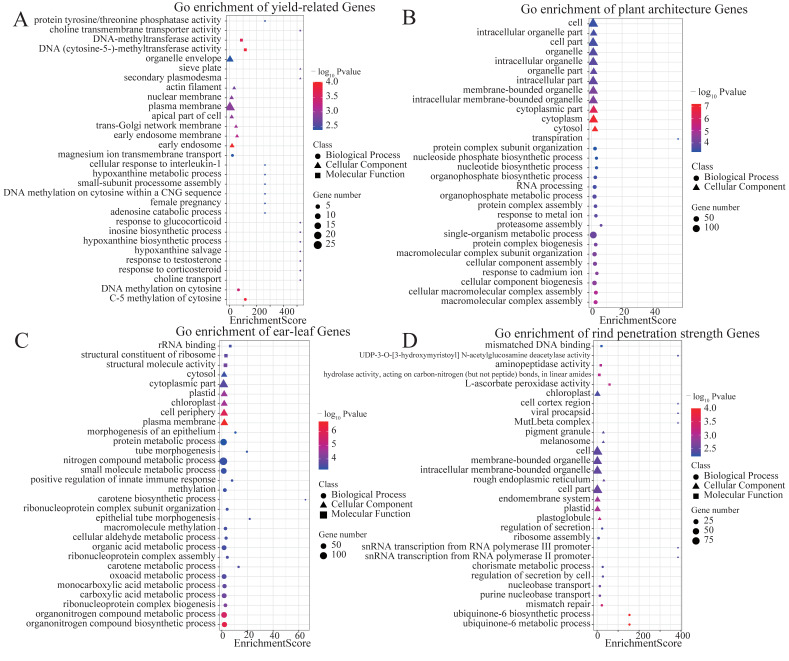
GO enrichment analysis of candidate genes for positive QTNs identified with model KinADE. (**A**): yield-related traits (GY and GM), (**B**): plant-architecture-related traits (PH, EH, and TBN), (**C**): ear-leaf-related traits (ELL, ELW, and ELA), (**D**): stem-puncture-strength-related traits (RPR_TIAG and RPR_IUE).

**Figure 6 ijms-25-01190-f006:**
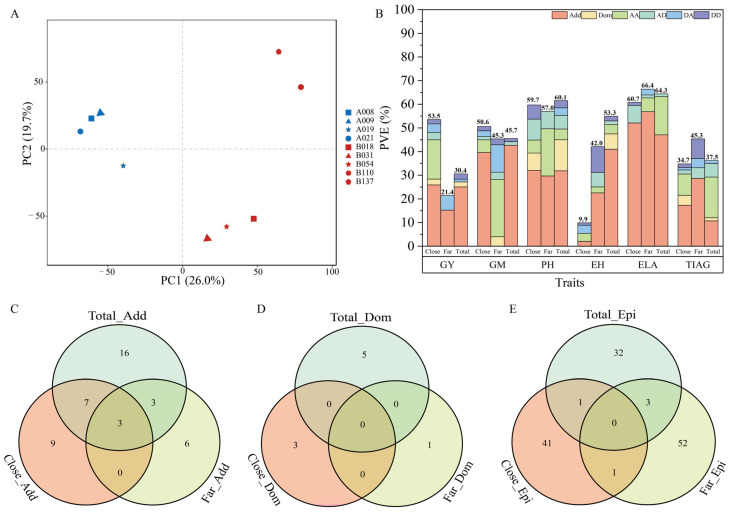
Influence of genetic distance among core test lines on mapping results. (**A**): Genetic relationship among those core test lines through the first two PCs of principle coordinate analysis, (**B**): Bar plot of phenotypic variation explained by QTNs identified in different populations, (**C**): Venn diagrams of significant additive effect QTNs identified in different populations, (**D**): Venn diagrams of significant dominance effect QTNs identified in different populations, (**E**): Venn diagrams of significant epistatic effect QTNs identified in different populations.

**Figure 7 ijms-25-01190-f007:**
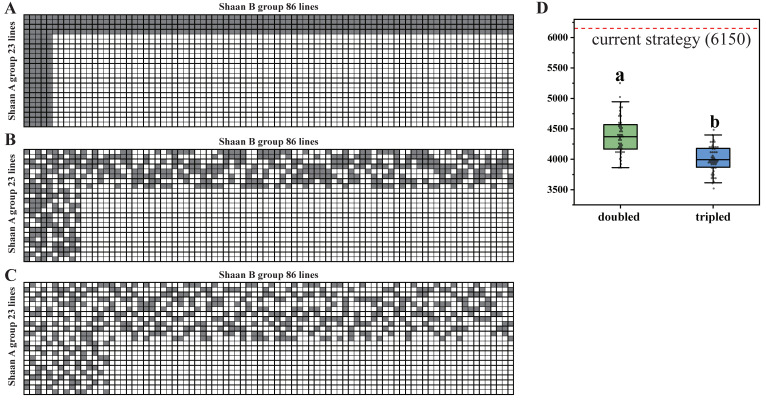
Mating strategies of different scale core tester lines for a typical breeder population. (**A**) Current mating strategy of four and five core tester lines from the Shaan A and Shaan B groups, respectively. (**B**) The first new strategy of doubling the core tester lines for both groups with the same number of hybrids. (**C**) The second new strategy of tripling the core tester lines for both groups with the same number of hybrids. (**D**) Number of one-genotype-missing-site (OGMS) in the F_1_ population under different breeding strategies; red line is the number of OGMSs in the current mating strategy, while the green and blue boxes represent the OGMS number distribution from 50 times resampling for the doubled and tripled strategy, respectively. Different letters show the significance level at α = 0.05.

**Figure 8 ijms-25-01190-f008:**
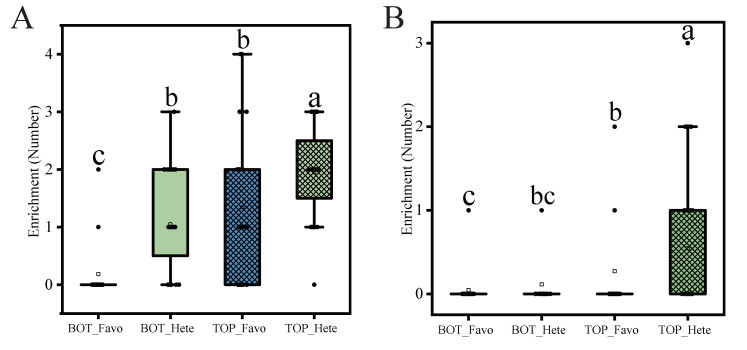
Number of favorable genotypes (Favo) and heterozygous genotypes (Hete) in the top (TOP) or bottom (BOT) 10% of the hybrid population for grain yield (GY). (**A**) The enrichment numbers of QTNs with additive and dominance effects. (**B**) The enrichment numbers of QTNs with epistatic effects. Different letters show the significance level at α = 0.05.

**Figure 9 ijms-25-01190-f009:**
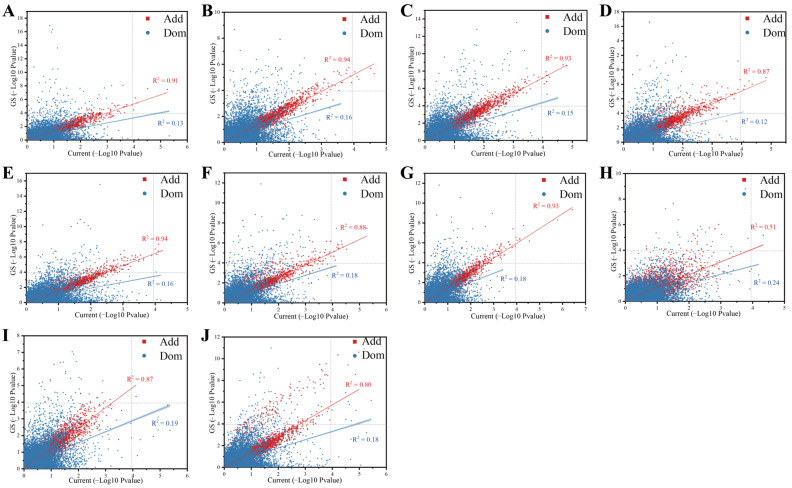
The correlation of significance of the SNPs between the predicted population and the current population. (**A**) GY, (**B**) GM, (**C**) PH, (**D**) EH, (**E**) ELL, (**F**) ELW, (**G**) ELA, (**H**) RPR_TIAG, (**I**) RPR_IUE, (**J**) TBN. The R^2^ of red and blue represents the determinative value between the GS strategy and the current population for the Add and Dom effects, respectively.

**Table 1 ijms-25-01190-t001:** Significant additive and dominance effect loci based on model KinADE.

Trait	SNP	Chr	Position	Genotype	Mode *	Effect	−log_10_P	PVE (%)
GY	*gySNP1*	1	215245696	T/A	Add	−0.5	4.0	6.7
	*gySNP2*	1	73916274	G/A	Add/Dom	0.92	5.3	2.1
	*gySNP3*	3	221920034	C/G	Add	−0.48	4.1	5.7
	*gySNP4*	5	209076386	G/T	Add	−0.56	4.5	12.6
	Sum							27.1
GM	*gmSNP1*	2	1263421	A/G	Add	0.48	4.2	7.5
	*gmSNP2*	5	15548317	C/T	Add	−0.91	4.6	2.4
	*gmSNP3*	7	157333857	C/T	Add	−0.77	4.5	12.6
	*gmSNP4*	8	129615225	G/A	Add	0.58	4.0	18.7
	*gmSNP5*	9	152684176	G/A	Add	−0.51	4.1	1.6
	Sum							42.8
PH	*phSNP1*	5	208463873	C/G	Add	9.69	4.7	9.6
	*phSNP2*	5	215535364	A/G	Add	8.39	4.0	13.2
	*phSNP3*	6	124049465	C/G	Add	9.64	4.1	8.6
	*phSNP4*	7	176596934	A/G	Add	−11.57	4.2	2.3
	*phSNP5*	8	128340728	C/A	Add/Dom	18.63	4.8	3.4
	*phSNP6*	10	147469999	C/T	Add/Dom	10.77	4.2	10.6
	Sum							47.7
EH	*ehSNP1*	3	291054	G/A	Add	−5.03	4.1	14.2
	*ehSNP2*	6	132789720	A/G	Add	−7.04	4.4	8.8
	*ehSNP3*	7	180279662	G/A	Add	−6.88	4.8	14.9
	*ehSNP4*	9	71431208	C/A	Add/Dom	10.61	4.3	6.5
	*ehSNP5*	9	79091347	C/G	Add/Dom	−10.57	4.1	1.7
	*ehSNP6*	9	99136257	C/T	Add/Dom	−7.2	3.9	1.4
	Sum							47.5
ELL	*ellSNP1*	1	254318451	G/A	Dom	2.04	4.0	2.2
	*ellSNP2*	6	118971410	C/T	Add/Dom	3.5	4.1	5.4
	*ellSNP3*	8	128340728	C/A	Add/Dom	4.97	4.0	9.6
	Sum							17.2
ELW	*elwSNP1*	2	21687575	T/A	Dom	−0.2	4.1	1.3
	*elwSNP2*	3	159442050	A/G	Add	0.45	4.4	8.2
	*elwSNP3*	3	229142753	G/C	Add	−0.4	5.3	2.5
	*elwSNP4*	5	196015158	C/T	Add/Dom	0.48	4.3	1.1
	*elwSNP5*	5	212255011	G/A	Add	−0.36	4.1	7.6
	*elwSNP6*	6	167994782	G/A	Add	−0.29	4.1	22.8
	*elwSNP7*	10	107630915	T/C	Add	0.41	4.5	15.6
	Sum							59.1
ELA	*elaSNP1*	2	195895775	G/T	Add/Dom	43.84	4.1	2.1
	*elaSNP2*	3	163190634	C/T	Add	−20.65	3.9	9.5
	*elaSNP3*	6	167994769	T/C	Add	−29.02	6.4	20.1
	*elaSNP4*	7	140317639	G/A	Add	26.56	4.0	10.6
	*elaSNP5*	7	142451671	G/A	Add	23.51	4.1	1.1
	*elaSNP6*	9	142662042	C/G	Add	23.81	4.0	3.6
	Sum							47.0
RPR_TIAG	*tiagSNP1*	1	244211691	T/A	Dom	2.59	4.2	1.3
	*tiagSNP2*	6	168202087	G/A	Add	3.2	4.3	4.2
	*tiagSNP3*	7	1403785	G/T	Add	−1.86	4.0	6.5
	Sum							12.0
RPR_IUE	*iueSNP1*	1	251572876	A/T	Add/Dom	1.72	4.0	3.6
	*iueSNP2*	6	157939881	T/C	Dom	−1.7	4.9	7.6
	*iueSNP3*	9	140917978	T/C	Add/Dom	2.49	4.2	8.1
	Sum							19.3
TBN	*tbnSNP1*	2	45027887	G/A	Add/Dom	1.05	4.7	1.4
	*tbnSNP2*	2	51045430	A/C	Add/Dom	1.24	5.1	6.0
	*tbnSNP3*	4	186665498	G/A	Add/Dom	−1.06	5.4	3.1
	*tbnSNP4*	4	189101304	A/T	Add/Dom	−0.98	5.3	2.2
	*tbnSNP5*	5	201203454	C/T	Dom	0.62	4.5	5.6
	*tbnSNP6*	6	134143465	A/T	Dom	−0.6	4.0	2.7
	Sum							21.0

*, Add: additive effect, Dom: dominance effect.

## Data Availability

The data presented in this study are available on request from the corresponding author.

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
