# Peer review of "Genome-Wide Association Study for Maize Hybrid Performance in a Typical Breeder Population"

_ijms, 2024, doi:10.3390/ijms25021190_

Round 1
Reviewer 1 Report
Comments and Suggestions for Authors
Dear Authors,
I have carefully reviewed the manuscript titled "Genome-Wide Association Study for Maize Hybrid Performance in A Typical Breeder Population" and would like to commend the authors for their valuable and novel contributions to the field of maize breeding and genetics.
The manuscript addresses a critical aspect of contemporary agriculture by delving into the genetic underpinnings of maize hybrid performance. The identification of significant SNPs, epistatic effects and the discovery of candidate genes, especially Zm00001d044211, provide valuable insights for enhancing yield and meeting global food demands.
The proposed novel mating strategy is an important practical implication, offering potential efficiency gains in genome-wide association studies. Such strategies are crucial for advancing plant breeding practices, ensuring food security and meeting the challenges posed by a growing global population.
However, I would like to draw attention to the need for a closer review of the English language in the manuscript. While the overall quality of writing is good, there may be some minor grammatical errors that an expert language editor could help identify and rectify. This step is crucial to ensure the clarity and professionalism of the manuscript.
I recommend that the authors engage the services of a language expert or professional editor to perform a detailed language check. This will help enhance the readability and overall presentation of their work.
Once these minor language issues are addressed, I believe the paper will be well-suited for publication and will make a meaningful contribution to the scientific community.
Thank you for considering my feedback and I look forward to seeing the refined version of this promising manuscript.
Author Response
Dear Authors,
I have carefully reviewed the manuscript titled "Genome-Wide Association Study for Maize Hybrid Performance in A Typical Breeder Population" and would like to commend the authors for their valuable and novel contributions to the field of maize breeding and genetics.
The manuscript addresses a critical aspect of contemporary agriculture by delving into the genetic underpinnings of maize hybrid performance. The identification of significant SNPs, epistatic effects and the discovery of candidate genes, especially Zm00001d044211, provide valuable insights for enhancing yield and meeting global food demands.
The proposed novel mating strategy is an important practical implication, offering potential efficiency gains in genome-wide association studies. Such strategies are crucial for advancing plant breeding practices, ensuring food security and meeting the challenges posed by a growing global population.
However, I would like to draw attention to the need for a closer review of the English language in the manuscript. While the overall quality of writing is good, there may be some minor grammatical errors that an expert language editor could help identify and rectify. This step is crucial to ensure the clarity and professionalism of the manuscript.
I recommend that the authors engage the services of a language expert or professional editor to perform a detailed language check. This will help enhance the readability and overall presentation of their work.
Once these minor language issues are addressed, I believe the paper will be well-suited for publication and will make a meaningful contribution to the scientific community.
Thank you for considering my feedback and I look forward to seeing the refined version of this promising manuscript.
Response:
Thank you for your positive comments and valuable suggestion!
Regarding writing and grammar issues, we revised our manuscript according your suggestion. Please refer to the track mark of modifications in the revised version.

Reviewer 2 Report
Comments and Suggestions for Authors
Please see the attachment.

Comments on the Quality of English Language
OK
Author Response
Reviewer2:
Firstly, thank you for your comments and constructive advices! We revised our manuscript according your suggestions. Responses and modifications are presented as follows:
Summary
Line 28 – Why? What conclusion supports this statement? Where does this fit into molecular biology and where does it fit into reproduction?
Response:
We diagrammed the multi-omic functions, characters, and GO enrichment of some important or whole candidate genes for those positive QTNs (Figure 4, 5, and S5). And we illustrated the allele frequency distribution and favorable genotypes expression for those positive QTNs (Figure 2, 3, S4, S6, and S7). We believe above results are valuable for further molecular biology research and breeding practice. Maybe your feeling is right, so we changed the “insights” to “basis”, please refer to the modification in Line 28-29 in the revised version.
Introduction
The authors report that researchers discovered crucial effects on the quantitative genetic architecture of corn.
OK! But why these studies? I believe that what is missing in the introduction is that the authors
highlight the gains from corn cultivation such as disease resistance, productivity gains,...
Show the reader the gains...
Response:
Thank you for your constructive suggestion! We inserted one sentence as “Report showed that genetic gain contributions to maize yield increases during 1930 to 2011 were 79% in the target production environment” in Line 35-36.
Materials and Method
https://www.mdpi.com/authors/layout#_bookmark57
2.1. Types of articles
2.1.1. Article:
The structure must include details of Abstract, Keywords, Introduction, Materials and Methods,
Results, Discussion and Conclusions (optional).
A doubt. On the magazine's website, in the guide, there is a sequence to be followed (Materials and methods come before results). However, I noticed that the articles published in this journal follow the order of results before Materials and methods.
Should authors follow the order described in the guide or the order of published articles?
Response: In reality, manuscripts in IJMS are typically formatted in the order of "Introduction," "Results," "Discussion and Conclusion," and "Materials and Methods." The following is a screenshot of the recommended submission format provided by the official IJMS journal during submission
Results
line 103 - 107 This part of the text should be in Materials and methods.
However, each trained attribute must be better detailed as it is evaluated. For example: cascade penetrameter resistance. What equipment was used? what unit of measurement?
Response:
Thank you for your suggestions! We placed such content at the beginning of the Results section to provide readers with a better understanding of the abbreviations representing various shapes that appear later. According to your suggestion, we provide units after the abbreviations of each trait. In order to simplify the manuscript, the detailed determination methods for each trait, we already set the reference [30] at the end of section “4.1”.
Figures 2, 3 and 5 have a very small font size.
Response:We enlarged the font size in Figure 2, 3, and 5.
Figure 4 is blurred. The words are not clear.
Response:We adjusted Figure 4 to enhance the clarity of the text within the figure as much as possible.
Discussion It's deep work. Fantastic! The authors addressed the results and discussion on the topic very well.
However, I felt that the authors could address a little discussion focused on rural producers. What quantitative impact will the results bring to the rural producer? What is the impact on the final productivity of corn? The discussion should also be directed to those on the farm
Response:While the discussion you raised regarding maize production is certainly important, it might not align closely with the focus of our study, research on maize planting policies or cultivation practices may be more suitable for addressing this aspect. Alternatively, our study primarily aids breeding efforts for breeders, while the gap between variety development and actual production is substantial. Considering the broad impact on rural areas and farmers is a complex and distant consideration for us.
Thank you very much!

Reviewer 3 Report
Comments and Suggestions for Authors
The manuscript is well-written and has scientific significance. The results are explained well and thoroughly discussed.
Minor queries:
1. Some previous studies (19-23) also employed the same strategy for identifying QTLs. How your study is diffeent from earlier studies. What gaps have been addressed in your work?
2. Why the expression data of other genes related to cell division, growth and differentiation, that may have role in yield development, is not checked in your transcriptome data for better understanding the regulation of grain yield.
3. Follow up of your study in terms of marker utilization in hybrid breeding may be discussed.
4. It would be much better if authors give a list of hybrids and parents as supplemetary file.
Author Response
Reviewer3:
The manuscript is well-written and has scientific significance. The results are explained well and thoroughly discussed.
Thanks for your positive comments!
Minor queries:
- Some previous studies (19-23) also employed the same strategy for identifying QTLs. How your study is different from earlier studies. What gaps have been addressed in your work?
Response:Thank you for your important suggestion! We added more sentences in Line 86-89 in the revised version.
- Why the expression data of other genes related to cell division, growth and differentiation, that may have role in yield development, is not checked in your transcriptome data for better understanding the regulation of grain yield.
Response:Your insight is well-founded. In our analysis, we also observed another gene, Zm00001d044212, in proximity to Zm00001d044211. This is a candidate gene encoding an auxin-induced protein, exhibiting consistently high expression levels in grains between 21-28 days. Unfortunately, the expression of Zm00001d044212 did not display a clear pattern across different materials, suggesting that while it may be associated with yield, it may not be the primary gene contributing to the yield variations in our study population. Regarding other genes related to cell division, growth, and differentiation, our GWAS did not identify similar genes. Specifically, transcriptomic data served as supplementary validation for our loci of interest rather than being the primary focus of our study.
- Follow up of your study in terms of marker utilization in hybrid breeding may be discussed.
Response:In Section 3.5 of the discussion, we explored the differences in marker enrichment between favorable and unfavorable combinations in yield traits. In Section 3.8, we screened potential materials for each trait. We believe these analyses reflect the utilization of these markers in hybrid breeding. As for a more in-depth study and discussion of these markers and candidate genes, some of these aspects are part of ongoing work in our laboratory by other researchers. Many thanks for your suggestion!
4. It would be much better if authors give a list of hybrids and parents as supplemetary file.
Response:Thank you for your constructive suggestion! Frankly, it already listed as Table S3 in our relevant paper [Ref.30]:Genome-wide prediction in a hybrid maize population adapted to Northwest China - ScienceDirect

Round 2
Reviewer 2 Report
Comments and Suggestions for Authors
The authors mainly improved the quality of the figures.
The manuscript became clearer with the change in figures and errors in the text were corrected.
The work is very well written and deserves publication!
I suggest accepting!
Comments on the Quality of English Language
ok